# The Very Long COVID: Persistence of Symptoms after 12–18 Months from the Onset of Infection and Hospitalization

**DOI:** 10.3390/jcm12051915

**Published:** 2023-02-28

**Authors:** Marco Ranucci, Ekaterina Baryshnikova, Martina Anguissola, Sara Pugliese, Luca Ranucci, Mara Falco, Lorenzo Menicanti

**Affiliations:** 1Department of Cardiovascular Anesthesia and Intensive Care, IRCCS Policlinico San Donato, 20097 Milan, Italy; 2Department of Radiology, Koelliker Hospital, 10134 Turin, Italy; 3Scientific Directorate, IRCCS Policlinico San Donato, 20097 Milan, Italy

**Keywords:** COVID-19, long COVID, post-acute, long follow-up, persistent symptoms, major physical symptoms, major neurological symptoms

## Abstract

According to the World Health Organization’s definition, long COVID is the persistence or development of new symptoms 3 months after the initial infection. Various conditions have been explored in studies with up to one-year follow-up but very few looked further. This prospective cohort study addresses the presence of a wide spectrum of symptoms in 121 patients hospitalized during the acute phase of COVID-19 infection, and the association between factors related to the acute phase of the disease and the presence of residual symptoms after one year or longer from hospitalization. The main results are as follows: (i) post-COVID symptoms persist in up to 60% of the patient population at a mean follow-up of 17 months; (ii) the most frequent symptoms are fatigue and dyspnea, but neuropsychological disturbances persist in about 30% of the patients (iii) when corrected for the duration of follow-up with a freedom-from-event analysis; only complete (2 doses) vaccination at the time of hospital admission remained independently associated with persistence of the major physical symptoms, while vaccination and previous neuropsychological symptoms remained independently associated with persistence of major neuropsychological symptoms.

## 1. Introduction

The World Health Organization’s definition of long COVID is the continuation or development of new symptoms 3 months after the initial SARS-CoV-2 infection, with these symptoms lasting for at least 2 months with no other explanation [1]. Common symptoms of long COVID can include fatigue, shortness of breath, and cognitive dysfunction. Over 200 different symptoms have been reported that can have an impact on everyday functioning [1]. A post-acute COVID-19 study from the USA [2] included patients hospitalized due to COVID-19 and re-evaluated after 60 days from discharge. Within this period of time, 6.7% of the patients died, 15.1% required re-admission, and 32.6% reported persistent symptoms. The most common symptoms were dyspnea while walking up the stairs, cough, and loss of taste and/or smell. Other studies extended the observation period up to 3–4 months after hospital admission and found similar results [3,4,5,6,7,8], with approximately 30% of the patient population reporting fatigue, dyspnea, psychological distress, anxiety, depression, concentration, and sleep abnormalities.

There are studies addressing mortality [9], cardiovascular risk [10], pulmonary function [11], and neuropsychological changes [12] at one year or longer after COVID-19 infection.

However, few studies extended the window of observation longer than 12 months. The present study addresses the persistence of a wide spectrum of symptoms after hospital discharge of COVID-19 patients, and the association between factors related to the acute phase of the disease and the presence of residual symptoms after one year or longer from hospitalization.

## 2. Materials and Methods

This is a single-center prospective cohort study conducted at the IRCCS San Donato, a Clinical Research Hospital partially funded by the Italian Ministry of Health. The Local Ethics Committee (San Raffaele Hospital) approved the experimental design on 3 March 2022, registry number 28/INT/2022. All the patients gave written informed consent. The study has been financed by a grant from the Italian Ministry of Health for the research projects of the Cardiac Network of the Italian IRCCS (Clinical Research Hospitals). The eligible patient population was represented by subjects hospitalized at our institution with a diagnosis of COVID-19 infection between January 2021 and July 2022. The planned patient population was 100 patients. The primary endpoint was the persistence of major physical and neuropsychological symptoms from 3 up to 12–18 months from the hospital discharge, and the assessment of factors associated with the time-related freedom from residual symptomatology. 

### 2.1. Patient Population and Study Procedures

The patients were recruited through an initial telephone contact; those who were reachable and agreed to participate received a date for the study procedure at our hospital. The telephone calls started in April 2022 and ended on 15 November 2022. The first patient hospital admission date was 8 January 2021, and the last hospital admission date was 4 July 2022. The first follow-up visit date was 12 April 2022, and the last follow-up visit was 21 November 2022. The recruitment flow is shown in Figure 1. The final patient population comprised 121 subjects. The study has three work packages. Work package 1 is a clinical assessment of the patient, comprehensive of the parameters at the time of hospital admission in the acute phase, main laboratory data, and investigation of the presence of residual symptomatology linked to COVID-19. Work package 2 is an evaluation of the coagulation profile of the patient, and work package 3 includes a proteomic assessment of the patient. The present report deals with the results of work package 1.

### 2.2. Data Collection and Definitions

Data collection was based on (i) the retrieval of the relevant data from the original patient’s files and (ii) a personal interview conducted in a hospital office by dedicated biologists and a medical doctor. 

The following items regarding the acute phase hospitalization were collected: demographics (with age classes ≤50 years, 51–60 years, 61–70 years, 71–80 years, and >80 years); disease severity (mild: no oxygen therapy; moderate: nasal oxygen or oxygen mask; and severe: non-invasive or invasive ventilation), hospital stay, the unit of admission, and vaccination (2 doses) at the time of hospital admission; co-morbidities: obesity, hypertension, diabetes, history of coronary disease, heart failure, atrial fibrillation, chronic obstructive pulmonary disease, asthma, active cancer, chronic kidney failure, chronic liver failure, previous cerebrovascular accident, anxiety, or depression; therapy at the time of hospitalization; laboratory exams: peak fibrinogen levels, peak D-Dimer, peak platelet count, nadir platelet count, or nadir antithrombin.

A specific COVID-19 treatment was applied to all the patients, following the indications of the Italian Drug Agency: a prophylactic dose of low molecular weight heparin; low-dose steroids; and remdesivir in selected patients (within 7 days from the onset of COVID-19 symptoms; a moderate degree of severity and at least one risk factor for progression to a severe form).

Follow-up items included follow-up duration; any symptom after discharge; work capacity reduced; fatigue, fever, cough, or dyspnea (these last four items combined as “Major physical symptoms”—MPS—adjudicated in the presence of one or more symptoms); and chest pain, palpitations, headache, sleep disturbances, anxiety, depression (new symptoms starting or worsening during hospitalization), memory dysfunction, brain fog, (these last four items combined as “Major neuropsychological symptoms”—MNS—adjudicated in presence of one or more symptoms), paresthesias, muscle pain, joint pain, and sensorial deficit. For each symptom or combination of symptoms, there was a distinction between resolved and ongoing status. Data were collected in an electronic platform (Research Electronic Data Capture—RedCAP).

### 2.3. Statistical Analysis

Data are shown as number (percentage), mean (standard deviation) or median (interquartile range) as appropriate. Differences between categorical variables were assessed using Pearson’s chi-square, while differences in continuous variables were explored with Student’s *t*-test (normally distributed variables) or a non-parametric test (non-normally distributed variables). Survival curves were applied in univariate (Kaplan–Meier with log-rank test) and multivariable (Cox regression with hazard ratios and 95% confidence intervals) analyses. For the statistical calculations and graphical support data were exported from RedCAP into statistical packages (SPSS 20.0, IBM, Chicago, IL, USA and MedCalc, MedCalc Software, Ostend, Belgium). For all tests, a *p* < 0.05 was considered significant.

## 3. Results

The general characteristics of the patient population are shown in Table 1, according to the severity of the disease in its acute phase. The body mass index (BMI) was significantly higher for increasing degrees of severity as well as the length of hospital stay. A previous smoking habit was significantly less frequent in those with a mild degree of severity, and patients with a severe pattern of disease had significantly higher peak fibrinogen values during the acute phase.

The data collected at follow-up are reported in Table 2 for the whole patient population and separately for the different degrees of severity of the disease during the acute phase. The follow-up period significantly differed, with a shorter follow-up for patients with a mild severity. Overall, 96% of the patients reported one or more symptoms from the hospital discharge to follow-up; however, this rate was significantly lower (79%) in patients with a mild pattern of the disease. MPS were reported as still present at the time of follow-up by 61% of the patient population, again with a significant lower rate (37%) in patients who experienced a mild pattern of disease. It is of notice that this patient population reported a significantly higher rate of fever; however, it was resolved at the time of follow-up. Among symptoms still present at the time of follow-up, fatigue was reported by 50% of the patients and dyspnea by 42%, followed by joint pain (35%), memory dysfunction (34%), sleep disturbances, muscle pain (27%), anxiety, brain fog and paresthesias (20%), and depression (18%). MPS were significantly more frequent for an increasing severity of the disease during the acute phase, and this particularly applied to ongoing dyspnea, whereas the other component of the MPS (fatigue and cough) did not differ for different degrees of the severity of the disease. The MNS rate was not significantly different for different degrees of the severity of the disease.

The analysis of the determinants of ongoing MPS and MNS was based on survival curves, given the different follow-up times between groups. Figure 2 reports the persistence of MPS in the general patient population. Starting with 95% of persistence after 3 months from hospital discharge, MPS remained present in 82% of the patients after 1 year, and 45% of the patients after 18 months, reaching about 10% only after 20 months. With a Kaplan–Meier analysis with log-rank test, factors associated with the freedom from MPS at a level of *p* < 0.1 were age class (with a faster resolution of symptoms for patients ≥ 70 years, *p* = 0.098) and vaccination, with vaccinated patients having a hazard ratio for the persistence of symptoms of 0.305 (95% confidence interval 0.164–0.568, *p* = 0.001) with respect to non-vaccinated patients. No other factor demonstrated an association with the persistence of MPS (Appendix A). Figure 3 reports the freedom from the persistence of MPS in vaccinated vs. non-vaccinated patients. After 1 year, 92% of the non-vaccinated patients still reported MPS vs. 50% of the vaccinated patients. After 18 months the percentages decreased to 50% and 22%, respectively.

A Cox regression analysis was applied to the persistence of MPS with vaccination, severity of disease, and age class as covariates. In this model, vaccination remained independently associated with the persistence of MPS (hazard ratio 0.309, 95% confidence interval 0.160–0.6, *p* = 0.001), whereas age class (hazard ratio 1.01, 95% confidence interval 0.843–1.212, *p* = 0.910) and severity of the disease (hazard ratio 0.970, 95% confidence interval 0.651–1.446, *p* = 0.880) lost significant association.

To investigate the intercorrelation between age class and vaccination, a sensitivity analysis was conducted (Table 3). Although without reaching statistical significance (*p* = 0.338); patients aged >80 years had a higher (40%) vaccination rate than patients ≤ 80 years (14 to 23%).

The persistence of MNS was significantly associated with vaccination (hazard ratio 0.206, 95% confidence interval 0.108–0.394, *p* = 0.001), ICU admission (hazard ratio 0.340, 95% confidence interval 0.184–0.628, *p* = 0.001), and presence of neurological symptoms (anxiety or depression) at the time of the acute phase (hazard ratio 2.87, 95% confidence interval 1.56–5.27, *p* = 0.001). The other factors showed no association with persistence of MNS (Appendix A). Once tested in a Cox regression multivariable analysis, the factors that remained associated with the persistence of MNS were vaccination (hazard ratio 0.205, 95% confidence interval 0.099–0.426, *p* = 0.001) and previous neurological symptoms (hazard ratio 3.42, 95% confidence interval 1.83–6.39, *p* = 0.001), while admission to the ICU lost significance (hazard ratio 0.59, 95% confidence interval 0.30–1.16, *p* = 0.126). Figure 4 shows the freedom from persistence of MNS in vaccinated and non-vaccinated patients, adjusted for previous neurological symptoms. At 1-year follow-up, vaccinated patients had MNS in 24% of the cases vs. 94% in non-vaccinated patients. At 18 months follow-up these rates remained stable for vaccinated patients and decreased to 32% in non-vaccinated patients.

Overall, at the time of the acute phase, 25 (20.7%) patients were vaccinated with two doses. After discharge, at the time of follow-up, 113 (93.4%) received a vaccination, with 9 patients (7.4%) receiving a total of one dose, 62 (51.2%) two doses, and 40 (33.1%) three doses. The type of vaccine was Pfizer in 77 (63.6%) patients, Moderna in 21 (17.4%) patients, Astra Zeneca in 7 (5.8%) patients, Janssen in 2 (1.7%) patients, and unknown in 6 patients. 

## 4. Discussion

The main results of our study are that (i) post-COVID symptoms persist in up to 60% of the patient population at a mean follow-up of 17 months; (ii) the most frequent symptoms are fatigue and dyspnea, but neuropsychological disturbances persist in about 30% of the patients (iii) when corrected for the duration of follow-up with a freedom-from-event analysis; only complete (2 doses) vaccination at the time of hospital admission remained independently associated with persistence of MPS, while vaccination and previous neuropsychological symptoms remained independently associated with persistence of MNS. Immediately after discharge, all the patients (vaccinated and non-vaccinated) showed residual symptoms, but they were compatible with the hospital discharge. The survival curves between vaccinated and non-vaccinated patients start diverging only after 3 months of follow-up.

Other studies investigated, with different experimental designs, the long-term persistence of symptoms after the hospital discharge of COVID-19 patients. Kalak and associates [13], in a series of 166 patients, 135 of whom had been admitted to a hospital during the acute phase, found residual weakness (21%), dyspnea (16%), and brain fog (7%) after 18 months from hospital discharge. These results are in line with our observation. The only predictors of dyspnea at 18 months follow-up were dyspnea during the acute phase and dexamethasone therapy, likely markers of the severity of the disease. Gutierrez-Canales and associates [14] addressed post-COVID symptoms in a series of patients (non-vaccinated) who did not require hospitalization during the acute phase. As expected, in these low-severity disease patients, those who were followed for 5 months or longer showed a lower rate of persistent symptoms (22%) with respect to our series. However, the frequency of each symptom followed our pattern, with fatigue, dyspnea, and neuropsychological disturbances being the most represented. A study performed in India [15] included 371 patients followed 6 months after the infection, with 22% of patients being admitted to a hospital during the acute phase. Again, the frequency of long COVID symptoms was lower (9%) than in our series. Of notice, patients who received two doses of vaccine before the infection had a higher probability of developing long COVID symptoms. The authors explained their findings as the result of a better survival. Our results show the opposite, with a time-adjusted probability of developing long COVID symptoms in vaccinated patients that is one-third for MPS and one-fifth for MNS with respect to non-vaccinated patients. In this regard, our study confirms what was reported by the United Kingdom Health Security Agency [16] that people vaccinated before COVID-19 infection are 50% less likely to develop long COVID symptoms 1 to 6 months after the infection. Our data show that this effect is prolonged and even more pronounced from 3 up to 18 months after the infection. The positive effects of vaccination in limiting the probability of developing long COVID symptoms were reported in a Spanish study on 681 patients (23% hospitalized during the acute phase) [17]. This study included only patients who developed long COVID symptoms after an unspecified period of time from the infection. Major symptoms like fatigue were significantly less frequent in vaccinated patients (odds ratio 0.19, 95% confidence interval 0.04–0.79). 

A comprehensive and multidisciplinary evaluation of patients hospitalized for COVID-19 infection was undertaken by Bellan and associates [18] one year after hospital discharge. Three hundred twenty-four patients received clinical investigation with lung function test, and a subgroup was tested for circulating cytokines. Patients admitted during the first wave had a persistence of symptoms of 41.1% vs. 31.2% in patients admitted during the third wave (*p* = 0.09) and showed a significantly (*p* = 0.02) lower diffusion lung for carbon monoxide. Although the authors did not address this point, no patient was vaccinated during the first wave in Italy, whereas many were during the third wave. Risk factors for the development of long COVID symptoms were gender female and previous neuropsychological symptoms (anxiety and depression), with no association with the severity of the disease. This finding is in agreement with our results. A Norwegian study [19] directly addressed the effects of vaccination on the presence of long COVID symptoms, in a population of 360 vaccinated and 1060 non-vaccinated patients. No significant differences were noticed with respect to permanent symptoms at 3–15 months. However, it was unspecified whether or not these patients were hospitalized during the acute phase.

Overall, the majority of the studies agree that long COVID symptoms are a frequent pattern, even if the window of observation is variable within and between studies. There is no consensus on the potential factors that may predict the onset of a long COVID pattern, and, namely, the severity of the disease in the acute phase, the age, and the role of vaccination. Conversely, the authors agree on the most common type of symptoms, which are fatigue, dyspnea, anxiety, and depression.

Our study differs from the existing ones and offers a different point of view. First of all, our patient population has been observed for a quite long period of time, up to 20 months after the hospital discharge, whereas the majority of the studies limit the follow-up to 6–12 months. Secondly, our patient population is homogeneous (only hospitalized patients) and single-center. This allowed us to review the patients’ files and retrieve objective information on a number of items, including vaccination state and laboratory exams. Third, and most important, our data have been retrieved through a direct interview of the patients rather than through telephone interviews or web-based questionnaires. This allowed a sounder and less subjective identification of the various symptoms. Finally, and differently from the existing studies, we could assess the freedom from long COVID symptoms using adequate statistical tools like actuarial curves with univariate and multivariable estimates of the role of various factors as determinants of MPS and MNS. This allowed us to discriminate the weight of each variable, and, in particular, we could demonstrate that, in our series, the apparent impact of the severity of the disease was blunted when analyzed in a model based on the follow-up time. The same happened for the age of the patients: the apparent paradox of a lower rate of residual symptoms in the elderly patients is explained by the higher rate of vaccinated patients in the age class >80 years. This is easily explainable considering that in Italy the first vaccinations were reserved for elderly people. Another possible explanation for the higher rate of long COVID pattern in young (<50 years) patients could be related to the fact that young and active subjects are probably more sensitive to a decrease in work capacity and endurance than elderly people. As a matter of fact, neuropsychological symptoms behave differently, with a trend toward a higher rate of persistence in elderly people.

The main result of our study pertains to the role of vaccination. In our series, patients vaccinated at the time of the acute phase had a one-third probability of developing major physical long COVID symptoms than non-vaccinated patients, and a one-fifth probability of developing major neuropsychological symptoms, independently from the severity of the disease. Therefore, the protective role of vaccination is not due to a lower severity of the acute phase, but to some alternative unknown mechanism that involves the immune system.

### Limitations

There are limitations to our study. The first one is that physical symptoms and diseases present at the time of hospitalization were retrieved from the patient’s files and are therefore reliable information; conversely, the pre-existence of symptoms comprising the MNS was explored in part from the patient’s files (anxiety and depression) and in part at the time of the follow-up interview, and it is possible that this last piece of information could be biased by the subjective interpretation and memory of the patient.

Even the severity of the symptoms is assessed based on subjective judgment and not on objective measures. The window of observation includes only 29 patients after 15 months of follow-up and therefore data at 18 months could be less reliable (full resolution of symptoms after 21 months is an extrapolation and cannot be proven). Finally, our patient population comprised only patients who, at the time of the acute phase, had symptoms requiring hospitalization. Hence, our results cannot be generalized to the general patient population and namely to those suffering an acute infection not requiring hospitalization. Even the single-center design limits the generalizability of our results, and a selection bias cannot be excluded.

In conclusion, long COVID represents an important sequela of the COVID-19 infection both in terms of physical and mental state. The wide diffusion of vaccination should however guarantee a significant containment of this pattern.

## Figures and Tables

**Figure 1 jcm-12-01915-f001:**
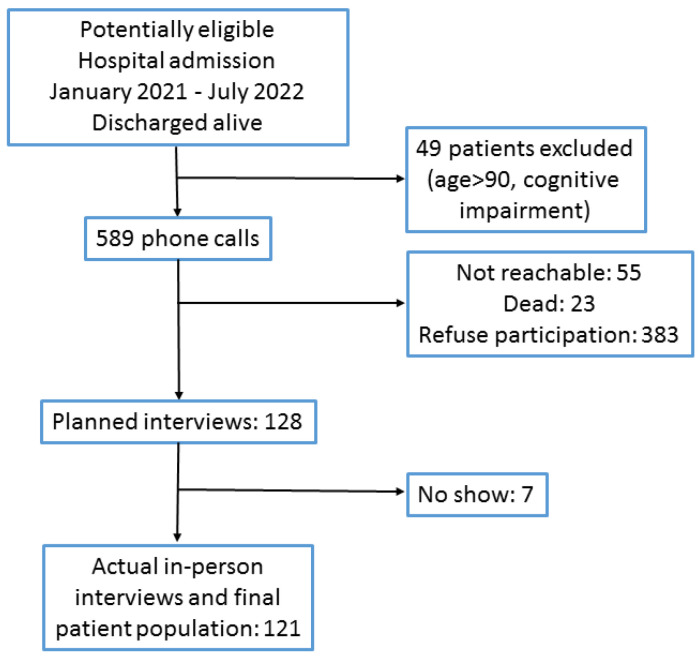
Patient screening and selection.

**Figure 2 jcm-12-01915-f002:**
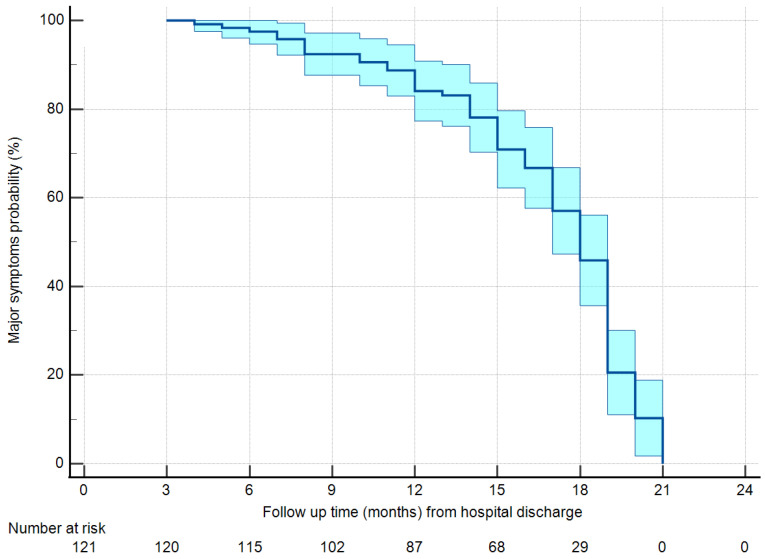
Freedom from major physical symptoms in the overall population.

**Figure 3 jcm-12-01915-f003:**
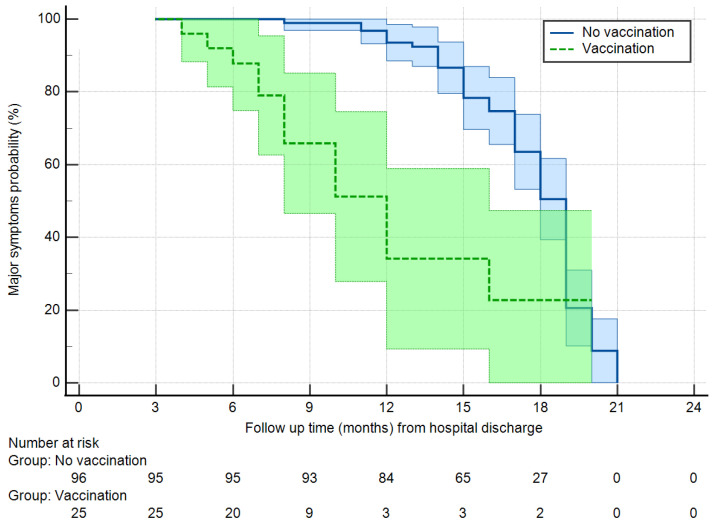
Freedom from major physical symptoms in vaccinated vs. non-vaccinated patients (Log-rank test, *p* = 0.001). Light blue and light green areas are 95% confidence intervals.

**Figure 4 jcm-12-01915-f004:**
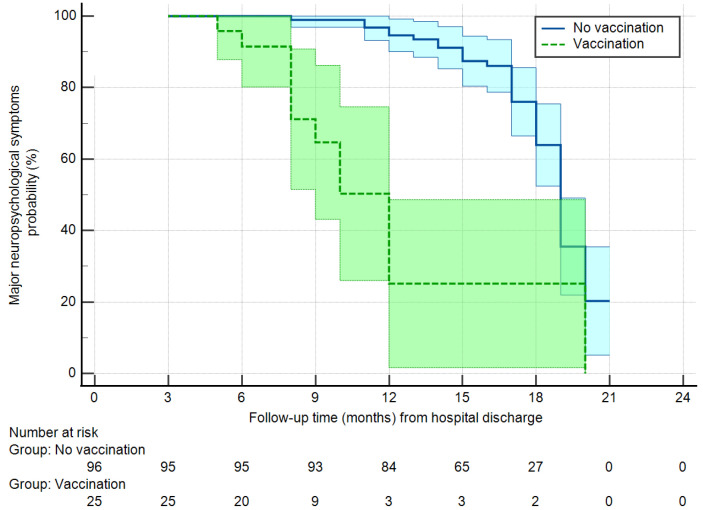
Freedom from major neuropsychological symptoms in vaccinated vs. non-vaccinated patients (Log-rank test, *p* = 0.001). Light blue and light green areas are 95% confidence intervals. Adjusted for previous neurological symptoms.

**Table 1 jcm-12-01915-t001:** Patient population (*N* = 121) details at hospital admission and during the acute phase of the disease, according to the severity of the disease.

Item	Mild	Moderate	Severe	*p*
	N = 19	N = 68	N = 34	
Age at hospital admission (years)	59.5 (15.3)	66.3 (11.6)	63.6 (14.6)	0.129
Gender male	12 (63.2)	46 (67.6)	22 (64.7)	0.916
Body mass index (kg/m^2^)	25.3 (4.1)	27.5 (6.0)	29.4 (6.0)	0.047
Hospital stay (days)	8 (6–18)	14 (10–20)	23 (16–32)	0.001
Unit of admission				
Ward	19 (100)	67 (98.5)	30 (91.2)	0.205
ICU	0 (0)	1 (1.5)	3 (8.8)	
Vaccination (at least 2 doses)	7 (36.8)	14 (19.1)	4 (11.8)	0.096
Obesity	3 (15.8)	14 (20.6)	11 (32.4)	0.294
Arterial hypertension	5 (26.3)	32 (47.1)	18 (52.9)	0.34
Diabetes	3 (15.8)	11 (16.2)	6 (17.6)	0.978
Coronaropathy	2 (10.5)	15 (22.1)	3 (8.8)	0.177
Heart failure	0 (0)	7 (10.3)	2 (5.9)	0.293
Smoking habit				0.008
No	11 (58)	33 (48.5)	15 (44.1)	
Previous	4 (21.1)	32 (47)	19 (56)	
Ongoing	4 (21.1)	3 (4.4)	0 (0)	
Atrial fibrillation	0 (0)	7 (10.3)	4 (11.8)	0.315
Active cancer previous 5 years	5 (26.3)	5 (7.4)	1 (2.9)	0.013
COPD	2 (10.5)	3 (4.4)	1 (2.9)	0.452
Chronic kidney failure	0 (0)	6 (8.8)	2 (5.9)	0.384
Previous CVA	2 (10.5)	1 (1.5)	3 (8.8)	0.13
Anxiety	3 (15.8)	9 (13.2)	7 (20.6)	0.629
Depression	3 (15.8)	6 (8.8)	5 (14.7)	0.56
Chronic liver failure	1 (5.3)	3 (4.4)	0 (0)	0.438

Data are mean (standard deviation), median (interquartile range), or number (%). CVA: cerebrovascular accident; COPD: chronic obstructive pulmonary disease; ICU: intensive care unit.

**Table 2 jcm-12-01915-t002:** Residual symptomatology after discharge from the hospital according to the severity of the acute phase.

Item	All	Mild	Moderate	Severe	*p*
		N = 121	N = 19	N = 68	N = 34
Follow-up time (months)	17 (12–18)	12 (8–17)	17 (12–18)	17 (14–19)	0.011
Symptoms after discharge	112 (95.6)	15 (78.9)	75 (95.6)	32 (94.1)	0.046
Major physical symptoms					
Resolved	38 (31.4)	12 (63.2)	26 (38.2)	9 (26.5)	
Ongoing	74 (61.2)	7 (36.8)	42 (61.8)	25 (73.5)	0.031
Work capacity reduced	39 (32.2)	4 (21.1)	21 (30.9)	14 (41.2)	0.303
Fever	5 (4.1)	3 (15.8)	2 (2.9)	0 (0)	0.016
Resolved	4 (3.3)	2 (10.5)	2 (2.9)	0 (0)	
Ongoing	1 (0.8)	1 (5.3)	0 (0)	0.348	
Fatigue	86 (71.1)	11 (57.9)	50 (73.5)	25 (73.5)	0.386
Resolved	26 (21.5)	4 (21.1)	15 (22.1)	7 (20.6)	
Ongoing	60 (49.6)	7 (36.8)	35 (51.5)	17 (50)	0.596
Cough	29 (24)	4 (21.1)	16 (23.5)	9 (26.5)	0.911
Resolved	9 (7.4)	1 (5.3)	6 (8.8)	2 (5.9)	
Ongoing	20 (16.5)	3 (15.8)	10 (14.7)	7 (20.6)	0.919
Dyspnea	59 (48.8)	1 (5.3)	37 (54.4)	21 (61.8)	0.001
Resolved	8 (6.6)	0 (0)	4 (5.9)	4 (11.8)	
Ongoing	51 (42.1)	1 (5.3)	33 (48.5)	17 (50)	0.001
Chest pain	12 (9.9)	1 (5.3)	7 (10.3)	4 (11.8)	0.845
Resolved	3 (2.5)	1 (5.3)	1 (1.5)	1 (2.9)	
Ongoing	9 (7.4)	0 (0)	6 (8.8)	3 (8.8)	0.62
Palpitations	13 (10.7)	1 (5.3)	8 (11.8)	4 (11.8)	0.215
Resolved	2 (1.7)	0 (0)	0 (0)	2 (5.9)	
Ongoing	11 (9.0)	1 (5.3)	8 (11.8)	2 (5.9)	0.17
Headache	14 (11.6)	2 (10.5)	7 (10.3)	5 (14.7)	0.211
Resolved	2(1.7)	0 (0)	2 (2.9)	0 (0)	
Ongoing	12 (9.9)	2 (10.5)	5 (7.4)	5 (14.7)	0.582
Major neuropsychological	58 (47.9)	7 (36.8)	35 (51.5)	16 (47.1)	0.525
symptoms					
Resolved	0 (0)	2 (5.3)	2 (2.9)		
Ongoing	54 (44.6)	7 (36.8)	33 (48.5)	14 (41.2)	0.592
Anxiety	26 (21.5)	3 (15.8)	16 (23.5)	7 (20.6)	0.76
Resolved	2 (1.7)	0 (0)	0 (0)	2 (5.9)	
Ongoing	24 (19.8)	3 (15.8)	16 (23.5)	5 (14.7)	0.177
Depression	24 (19.8)	2 (10.5)	14 (20.6)	8 (23.5)	0.509
Resolved	2 (1.7)	0 (0)	0 (0)	2 (0)	
Ongoing	22 (18.1)	2 (10.5)	14 (20.6)	6 (17.6)	0.183
Sleep disturbances	36 (29.8)	3 (15.8)	22 (32.4)	11 (32.4)	0.35
Resolved	3 (2.5)	1 (5.3)	1 (1.5)	1 (2.9)	
Ongoing	33 (27.3)	2 (10.5)	21 (30.9)	10 (29.4)	0.425
Memory dysfunction	46 (38)	6 (31.6)	26 (38.2)	14 (41.2)	0.787
Resolved	4 (3.3)	1 (5.3)	2 (2.9)	1 (2.9)	
Ongoing	42 (34.7)	5 (26.3)	24 (35.3)	13 (38.2)	0.916
Brain fog	29 (24)	4 (21.1)	17 (25)	8 (23.5)	0.936
Resolved	4 (3.3)	1 (5.3)	2 (2.9)	1 (2.9)	
Ongoing	25 (20.6)	3 (15.8)	15 (22.1)	7 (20.6)	0.966
Paresthesia	31 (25.6)	1 (5.3)	21 (30.9)	9 (25.5)	0.077
Resolved	6 (5.0)	0 (0)	3 (4.4)	3 (8.8)	
Ongoing	26 (20.6)	1 (5.3)	18 (26.5)	5 (14.7)	0.13
Muscle pain	39 (32.2)	6 (31.6)	22 (32.4)	11 (32.4)	0.998
Resolved	7 (5.8)	1 (5.3)	5 (7.4)	1 (2.9)	
Ongoing	32 (26.5)	5 (26.3)	17 (25)	10 (29.4)	0.919
Joint pain	45 (37.2)	5 (26.3)	27 (39.7)	13 (38.2)	0.559
Resolved	3 (2.5)	1 (5.3)	2 (2.9)	0 (0)	
Ongoing	42 (34.7)	4 (21.1)	25 (36.8)	13 (38.2)	0.533
Residual sensory deficit	48 (39.6)	3 (15.8)	30 (44.1)	15 (44.1)	0.068
Sight	19 (15.7)	1 (5.3)	8 (11.8)1	0 (29.4)	0.037
Smell	5 (4.1)	1 (5.3)	2 (2.9)	2 (5.9)	0.786
Hearing	12 (9.9)	1 (5.3)	10 (14.7)	1 (2.9)	0.543
Taste	12 (9.9)	0 (0)	10 (14.7)	2 (5.9)	0.653

Data are median (interquartile range) or number (%).

**Table 3 jcm-12-01915-t003:** Vaccination rate according to age class.

Age class	<50	50–60	61–70	70–80	>80	*p*
	N = 17	N = 26	N = 28	N = 35	N = 15	
Vaccination	4 (23.5)	5 (19.2)	5 (17.9)	5 (14.3)	6 (40)	0.338

Data are number (%).

## Data Availability

The original dataset supporting the findings of this study will be deposited in the public repository Zenodo after the publication of the paper and accessible upon a reasonable request to the corresponding author.

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
