# Peer review of "The Very Long COVID: Persistence of Symptoms after 12–18 Months from the Onset of Infection and Hospitalization"

_jcm, 2023, doi:10.3390/jcm12051915_

Round 1
Reviewer 1 Report
The article by Ranucci et al. looks at the persistence of symptoms 12-18 months after hospitalization for acute COVID-19. This is a useful study as most investigations into PASC are limited to 3-12 months following infection; however, there is evidence that symptoms can persist for significantly longer than one year. The authors also provide models for the decay of symptoms in vaccinated and unvaccinated hospitalized patients over a 21 month period and show that vaccination increases the rate of symptom decay in their population. This is a useful study that contributes to our understanding of symptom persistence following COVID-19 hospitalization; however, there are several issues that should be addressed:
1. In the methods it is not clear which of the follow-up items were assessed though patient records and which though the follow-up survey. Presumably, at the survey patients were asked about on-going symptoms, but how were previous (and resolved) symptoms assessed? If this was only captured during the survey ~17 months after infection, it is very possible that some resolved symptoms might not be recalled. Were these symptoms also captured in the medical records at time of discharge, which would at least provide a starting point for the range of symptoms, though some patients report certain symptoms only first appearing weeks after the resolution of the initial infection. These aspects need more discussion and clarification.
2. In the clusters of symptoms MPS and MNS, it should be made clear if this means having any one of the included symptoms or all of the included symptoms (presumably one).
3. In the patient population in Table 1 (or in the discussion of Table 1 data), it is important to mention if any of the other symptoms in Table 2 were present during hospital admission. This has implication when assessing the factors associated with persistence of MPS and MNS. For example, it appears that only anxiety and depression were investigated for association with persistence of MNS, but there are multiple other neurological symptoms/conditions that may be associated that are described in Table 2. In addition, most of the data in Table 1 implies pre-existing conditions from before hospitalization, even if they were recorded at admission. However, it is not clear if anxiety and depression represent pre-existing conditions or symptoms seen during acute infection only (presumably during infection only, but this should be made clear). Finally, what percentage of vaccinated/unvaccinated patients had symptoms after discharge. This is useful to include as from the survival curves it appears that vaccination did not decrease the percentage of the population with MPS or MNS at 3 months.
4. In Table 2 for All, the follow-up time is listed as 17 (12-18), whereas presumably it should be 17 (8-19), including the full range of mild/moderate/severe.
5. When assessing the factors associated with freedom from MPS and MNS, which factors were tested and what were their p values? Presumably, multiple factors were tested (e.g. gender, pre-existing health conditions, medications during hospitalization, etc), these should be listed as a table (or a supplemental table) with their p values or discussed in the text.
6. In the analysis of figure 4, how was the data adjusted for previous symptoms?
7. The article discussions should include the limitations of the study (e.g. only hospitalized patients, non-representative aspects of the patient population, data extrapolation only beyond 18 months, no assessment symptom severity, etc.). This is especially important with the survival curves and the prediction beyond the study period to 21 months, as the curves predict 100% resolution by 21 months, but there are reports of symptom persistence beyond this point and it is not known if there is more of a tail of symptom persistence in the study population. This is a very important point for the public messaging of the study, as from the data shown, full resolution by 21 months cannot be conclusively proven.
8. Critically, did anyone in the study receive a vaccine dose after hospital discharge, either the primary series for the unvaccinated or a third dose booster for the vaccination group? As vaccination of PASC patients can affect symptom severity/progression and over the ~17 month follow-up additional vaccination was highly possible, this needs to be mentioned and if anyone in the study group received a dose during the follow-up, this should be discussed and accounted for in the data analysis.
9. Finally, there are also a few small points: Line 313 and 315, permanence should be persistence; which vaccine patients received should be listed; and it would be helpful to mention the dominant variant in the region at the time of most infections in the cohort.
Author Response
The article by Ranucci et al. looks at the persistence of symptoms 12-18 months after hospitalization for acute COVID-19. This is a useful study as most investigations into PASC are limited to 3-12 months following infection; however, there is evidence that symptoms can persist for significantly longer than one year. The authors also provide models for the decay of symptoms in vaccinated and unvaccinated hospitalized patients over a 21 month period and show that vaccination increases the rate of symptom decay in their population. This is a useful study that contributes to our understanding of symptom persistence following COVID-19 hospitalization; however, there are several issues that should be addressed:
- In the methods it is not clear which of the follow-up items were assessed though patient records and which though the follow-up survey. Presumably, at the survey patients were asked about on-going symptoms, but how were previous (and resolved) symptoms assessed? If this was only captured during the survey ~17 months after infection, it is very possible that some resolved symptoms might not be recalled. Were these symptoms also captured in the medical records at time of discharge, which would at least provide a starting point for the range of symptoms, though some patients report certain symptoms only first appearing weeks after the resolution of the initial infection. These aspects need more discussion and clarification.
REPLY: we thank the Reviewer for raising this point. Actually, a piece of information was retrieved from the original files and laboratory data. For example, chronic renal disease adjudicated based on serum creatinine clearance, atrial fibrillation from ECG, obesity from demographics. Going more into the details of symptoms explored at follow-up, the MPS was adjudicated in presence of one of the following: fatigue, fever, cough, dyspnea. The presence of at least one of these symptoms at the time of hospital admission is sure, otherwise the patient would not be hospitalized (please consider that our series includes ONLY hospitalized patients). Things are of course different for neuropsychological symptoms. MNS was adjudicated at follow-up only if not present before hospitalization. The presence of at least one of the items composing the MNS at the time of hospitalization was retrieved by the patient’s files for the items “anxiety” and “depression”, and using the questionnaire at the time of follow-up visit. We acknowledge that this may be a source of bias, and we have now discussed this in the limitations.
- In the clusters of symptoms MPS and MNS, it should be made clear if this means having any one of the included symptoms or all of the included symptoms (presumably one).
REPLY: yes, at least one symptom is enough for adjudicating MPS/MNS. We have now included this in the Methodology and Definitions.
- In the patient population in Table 1 (or in the discussion of Table 1 data), it is important to mention if any of the other symptoms in Table 2 were present during hospital admission. This has implication when assessing the factors associated with persistence of MPS and MNS. For example, it appears that only anxiety and depression were investigated for association with persistence of MNS, but there are multiple other neurological symptoms/conditions that may be associated that are described in Table 2.
REPLY: we could not address additional symptoms in table 1, because they were not mentioned routinely in the patient’s files (i.e. sleep disturbances, memory dysfunction, brain fog….). These items were defined as “non-pre-existing” based on the patient interview at follow-up. Therefore, being this piece of information not derived from patient’s file, we could not include it in Table 1, nor analyze it for association with MPS/MNS. We added this in the limitations.
In addition, most of the data in Table 1 implies pre-existing conditions from before hospitalization, even if they were recorded at admission. However, it is not clear if anxiety and depression represent pre-existing conditions or symptoms seen during acute infection only (presumably during infection only, but this should be made clear).
REPLY: anxiety and depression were reported in the anamnesis, so they are to be considered pre-existing to the acute phase. For the adjudication of MNS, we relied on the patient interview, where the patients defined anxiety and depression beginning at the time of infection or worsened as a consequence of the infection and hospitalization. We have now clarified this.
Finally, what percentage of vaccinated/unvaccinated patients had symptoms after discharge. This is useful to include as from the survival curves it appears that vaccination did not decrease the percentage of the population with MPS or MNS at 3 months.
REPLY: actually, all the patients (vaccinated or non-vaccinated) were discharged with at least one physical /neuropsychological symptom (usually, fatigue, anxiety, sleep disturbances). This can be detected by the survival curves that start at 95%. The reason why all the patients were discharged with residual symptoms was that there was a huge pressure for admitting new patients to the hospital facilities. Of course, the patients were discharged when free from symptoms like fever or severe dyspnea, but were usually still suffering from fatigue, anxiety, sleep disturbances or other symptoms that were however suitable for a discharge home. Again, this concept has now been added to the Results and Discussion.
- In Table 2 for All, the follow-up time is listed as 17 (12-18), whereas presumably it should be 17 (8-19), including the full range of mild/moderate/severe.
REPLY: no, the range is correct for “All”. This is because (as per table legend) this is not a full range, but an interquartile range – from 25th to 75th percentile).
- When assessing the factors associated with freedom from MPS and MNS, which factors were tested and what were their p values? Presumably, multiple factors were tested (e.g. gender, pre-existing health conditions, medications during hospitalization, etc), these should be listed as a table (or a supplemental table) with their p values or discussed in the text.
REPLY: the Reviewer is right, we tested all factors listed in table 1, but only P values < 0.1 were considered, tested and reported. We have now added 2 supplementary tables with the P values for association with MPS and MNS of all the covariates.
- In the analysis of figure 4, how was the data adjusted for previous symptoms?
REPLY: figure 4 is adjusted for previous symptoms. The adjustment is described in the text, and a multivariable Cox regression analysis was used for adjustment (the usual technique is adjustment for median value of the adjustment covariate). We have now clarified this by adding “unadjusted – adjusted analysis” to the legends of figure 3 and 4.
- The article discussions should include the limitations of the study (e.g. only hospitalized patients, non-representative aspects of the patient population, data extrapolation only beyond 18 months, no assessment symptom severity, etc.). This is especially important with the survival curves and the prediction beyond the study period to 21 months, as the curves predict 100% resolution by 21 months, but there are reports of symptom persistence beyond this point and it is not known if there is more of a tail of symptom persistence in the study population. This is a very important point for the public messaging of the study, as from the data shown, full resolution by 21 months cannot be conclusively proven.
REPLY: we absolutely agree and we have now added a comprehensive “limitations” paragraph.
- Critically, did anyone in the study receive a vaccine dose after hospital discharge, either the primary series for the unvaccinated or a third dose booster for the vaccination group? As vaccination of PASC patients can affect symptom severity/progression and over the ~17 month follow-up additional vaccination was highly possible, this needs to be mentioned and if anyone in the study group received a dose during the follow-up, this should be discussed and accounted for in the data analysis.
REPLY: We have now expressed the vaccination rate and the doses received after hospital discharge and before the follow-up visit. Almost all patients were partially or fully vaccinated at the time of follow-up visit. We have now even indicated which kind of vaccine was used. However, it is our opinion that data on post-COVID vaccination cannot be analyzed as possible factors changing the persistence of MPS/MNS. As a matter of fact, those classified as “post-COVID vaccination” of course were not vaccinated at the time of hospitalization; therefore, it may apparently result that receiving a vaccination AFTER the acute phase worsens the symptoms. The real statistical point is that being vaccinated at the time of infection is protective against long-COVID symptoms, and this can mask the potential benefits of a late vaccination. The only possible comparison is between “vaccinated at any time” vs. “non vaccinated”, but we have only 8 patients in this last group.
- Finally, there are also a few small points: Line 313 and 315, permanence should be persistence; which vaccine patients received should be listed; and it would be helpful to mention the dominant variant in the region at the time of most infections in the cohort.
REPLY: we have corrected the mistake and indicated the type of vaccine utilized.
Reviewer 2 Report
Comments to authors
;
Ranucci et al. present an interesting prospective, single-center study of COVID-19 patients included from January 2021 to July of 2022, with a three levels of disease severity. Their main finding is the protective effect of vaccinations on long-covid symptoms. The patient population of 121 patients were followed for up to 20 months. The authors show a high persistence of symptoms even 12 months after acute disease.
The follow up period in this study is among the longest and the authors used compelling statistical analyses to investigate their results.
Although the authors present a very interesting paper, several major issues need to be addressed in order to improve quality and readability of the paper.
1- Methods
- Please clarify how many patient beds are available at the hospital? It seems that between March 2021 and September 2022 (over 18 months), only 638 patients were admitted, is that correct? If so, can you elaborate on the population number in the area served by the hospital? This directly affects the generalizability of the findings.
- line 52-52, the sentence is not clear and should be corrected.
- Lines 60-62 you mention patients that are eligible between January 2021 and July 2021, however, in Figure 1 different dates are mentioned, please clarify.
- It is not clear how many follow-up for each patient were there. Were there specific time frames : 3, 6, 9, 18 months etc…? Is so, how many follow-ups were done and at what intervals? Were all the follow-ups data?. Also, when was the primary phone call done? Just after discharge or several weeks/months after?
- When was the first patient admitted and the last one? What date was the last follow-up visit?
- How many patients were interviewed by biologists? And how many by physicians? How many staff members were involved in the interviews ?
- Fever is not a symptom strictly speaking. Was fever measured during the acute disease or only reported by patients?
- I would suggest changing arrhythmias to palpitations
- Why chest pain, headache, arrhythmias, joint pain, muscle pain sensorial deficit and paresthesia are not considered as part of the Major Physical symptoms? To note, most of these have higher incidence than fever which was included in MPS.
- Did the authors use any severity indices for any of the symptoms? And also please elaborate how the symptoms were reported, using open questions or standardized questionnaire ?
2- Table 1
The table is a bit crowded and should be simplified.
- Weight is not necessary i think since you use BMI
- did any patient get more than 2 doses of vaccine and if so how many?
-The therapy and laboratory exams section in the table can be omitted and presented in a supplement.
- The therapies mentioned do not include any of the recommended medications used during 2021 as a treatment for covid-19, like dexamethasone, enoxoparin, remdesivir … this should be discussed and also considered as one of the limitations of the study.
- line 182-183 was the fever reported or measured?
3- please clarify if any of the 121 were lost to follow up.
4- Discussion
- The authors did not mention any limitation in their discussion, i would suggest to add a paragraph listing all the limitations and an elaboration on each one.
- The single center designs limits the generalizability of the findings. There is an obvious selection bias in which patients who are still symptomatic are more likely to participate in the interviews than symptom-free individuals. Of the 589 phone calls done only 121 were included and 383 declined participation which significantly affects the generalizability of the findings.
- Medications, as mentioned before, do not meet the recommended treatments at the time of the study, which may affect the prevalence of symptoms.
- The authors did not mention if any of the patients had an infection prior to their hospitalization or during the follow-up period, this can have a major impact on the prevalence of symptoms.
- The authors should consider the effect of different strains of covid-19 were all the patients infected by the same strain?
Author Response
Ranucci et al. present an interesting prospective, single-center study of COVID-19 patients included from January 2021 to July of 2022, with a three levels of disease severity. Their main finding is the protective effect of vaccinations on long-covid symptoms. The patient population of 121 patients were followed for up to 20 months. The authors show a high persistence of symptoms even 12 months after acute disease.
The follow up period in this study is among the longest and the authors used compelling statistical analyses to investigate their results.
Although the authors present a very interesting paper, several major issues need to be addressed in order to improve quality and readability of the paper.
1- Methods
- Please clarify how many patient beds are available at the hospital? It seems that between March 2021 and September 2022 (over 18 months), only 638 patients were admitted, is that correct? If so, can you elaborate on the population number in the area served by the hospital? This directly affects the generalizability of the findings.
REPLY: we have 380 beds in our Hospital. Our Institution is basically a Cardiological Center that remained active as a Hub for cardiac patients during the second and third wave of COVID-19. This means that only a minority of the beds were dedicated to COVID patients. Additionally, a certain, not trivial, amount of patients died during hospitalization. So the total amount of patients admitted in our Hospital for COVID-19 was around 1,000. There were other hospitals, of course, acting in our area as “COVID-19 Hub”.
- line 52-52, the sentence is not clear and should be corrected.
REPLY: the sentence has been rephrased into “The present study addresses the persistence of a wide spectrum of symptoms after hospital discharge of COVID-19 patients, and the association between factors related to the acute phase of the disease and the presence of residual symptoms after one year or longer from hospitalization.”
- Lines 60-62 you mention patients that are eligible between January 2021 and July 2021, however, in Figure 1 different dates are mentioned, please clarify.
REPLY: The correct figures are those in text, we have now corrected Figure 1.
- It is not clear how many follow-up for each patient were there. Were there specific time frames : 3, 6, 9, 18 months etc…? Is so, how many follow-ups were done and at what intervals? Were all the follow-ups data?. Also, when was the primary phone call done? Just after discharge or several weeks/months after?
REPLY: this study was not planned at the time of patients hospitalization for the majority of the patients. It was approved in march 2022 and the telephone calls started in April 2022 and ended on November 21st, 2022. So, there were no pre-defined time frames for follow-up. The follow-up ranged from 3 months (patients hospitalized in July 2022 to 21 months. Hence, the primary (and only) phone call could be at variable intervals from hospital discharge. We have now included this in the Methodology.
- When was the first patient admitted and the last one? What date was the last follow-up visit?
REPLY: first patient hospital admission date was January 8th, 2021, last hospital admission July 4th, 2022. First follow-up visit date was April 12th, 2022, last follow-up visit was November 21st, 2022. We have now included this information in Methodology.
- How many patients were interviewed by biologists? And how many by physicians? How many staff members were involved in the interviews ?
REPLY: there was one physician interviewing the patients and 2 biologists. All the interviews were conducted by one of the biologists together with the physician. The biologists were in charge for RedCap platform data insertion and the procedures of Workflow 2 and 3 that are not part of the present article. The interview was performed based on a Questionnaire with a pre-determined set of questions and multiple choice responces.
- Fever is not a symptom strictly speaking. Was fever measured during the acute disease or only reported by patients?
REPLY: fever during acute disease was retrieved from patient’s files. Fever after discharge was mentioned by the patients at the follow-up interview
- I would suggest changing arrhythmias to palpitations
REPLY: agreed and done
- Why chest pain, headache, arrhythmias, joint pain, muscle pain sensorial deficit and paresthesia are not considered as part of the Major Physical symptoms? To note, most of these have higher incidence than fever which was included in MPS.
REPLY: thanks for this question. We could have included other symptoms within the MPS definition, but this was pre-defined before data analysis. Of notice, only 4 patients without MPS actually referred other symptoms.
- Did the authors use any severity indices for any of the symptoms? And also please elaborate how the symptoms were reported, using open questions or standardized questionnaire ?
REPLY: no, we did not use any index to assess the severity of the symptoms. This has now been included in the Limitations paragraph. Of course, many of the symptoms composing the MPS would require specific tests to assess the degree of severity (i.e cardiopulmonary exercise test for fatigue; pulmonary function tests for dyspnea…). Additionally, we are lacking a measure before the acute phase to quantify the residual damage. All this is outside the design and purpose of our study. About the questionnaire, we used a standardized questionnaire that is available on request.
2- Table 1
The table is a bit crowded and should be simplified.
- Weight is not necessary i think since you use BMI
REPLY: agreed, we now skipped the weight
- did any patient get more than 2 doses of vaccine and if so how many?
REPLY: we have now indicated vaccination details in the results
-The therapy and laboratory exams section in the table can be omitted and presented in a supplement.
REPLY: agreed and done.
- The therapies mentioned do not include any of the recommended medications used during 2021 as a treatment for covid-19, like dexamethasone, enoxoparin, remdesivir … this should be discussed and also considered as one of the limitations of the study.
REPLY: we thank the Reviewer for raising this point. Actually, we have only mentioned the therapies that were received before and during the hospitalization, and not those directly addressing the COVID-19.
We have now included the treatment followed by our series of patients, that followed the standards of care (as per the Italian Drug Agency indications) at the time of hospitalization: prophylactic dose of LMWH; low-dose steroids; and remdesivir in selected patients (within 7 days from the onset of COVID-19 symptoms; moderate degree of severity and at least one risk factor for progression to a severe form.
- line 182-183 was the fever reported or measured?
REPLY: fever was reported at the time of follow-up. Being resolved in the majority of the cases, a measure had no sense. Only one patient reported ongoing fever episodes, but not present at the time of the interview.
3- please clarify if any of the 121 were lost to follow up.
REPLY: figure 1 contains this information. The only follow-up is represented by the single interview: among those who accepted to be interviewed at our hospital, only 7 did not show. We are not planning at the moment other follow-up visits.
4- Discussion
- The authors did not mention any limitation in their discussion, i would suggest to add a paragraph listing all the limitations and an elaboration on each one.
REPLY: we have now added a limitation paragraph inclusive of all the limitations highlighted by this and the other Reviewer.
- The single center designs limits the generalizability of the findings. There is an obvious selection bias in which patients who are still symptomatic are more likely to participate in the interviews than symptom-free individuals. Of the 589 phone calls done only 121 were included and 383 declined participation which significantly affects the generalizability of the findings.
REPLY: we have now added the single center study to the limitations. About the selection bias quoted by this Reviewer, we are not totally in agreement. It could be that those who denied to be interviewed were those symptom-free, but it could be even the opposite: this is an elderly patient population, and all the patients had to move from home to the hospital so it could be that in presence of fatigue or dyspnea they decided not to participate. Willingness to participate could also be minor in patients suffering anxiety or depression. Finally, the patients accepted not only the interview, but even to receive a venipuncture for the blood samples required by Workpackages 2 and 3 (see in the text).
- Medications, as mentioned before, do not meet the recommended treatments at the time of the study, which may affect the prevalence of symptoms.
REPLY: as already highlighted, all the patients received the standard of care. We apologize for not quoting this in our first version, it is now in the Results section.
- The authors did not mention if any of the patients had an infection prior to their hospitalization or during the follow-up period, this can have a major impact on the prevalence of symptoms.
REPLY: sorry, this information is not retrievable in a reliable way.
- The authors should consider the effect of different strains of covid-19 were all the patients infected by the same strain?
REPLY: again, this information is lacking; and, sincerely, we are not convinced that could affect the results. At the time of the second/third wave, in Italy, the Omega variant was starting to spread, and lately the Delta.

Round 2
Reviewer 1 Report
The revisions and additional supplemental data have improved the manuscript and adequately addressed the previously-raised issues.
Reviewer 2 Report
Thnak you for the revision, i think the paper is much better in the present form